# In-Hand Gravitational Pivoting Using Tactile Sensing

**Jason Toskov**
Monash University
jtos0003@student.monash.edu

**Rhys Newbury**
Monash University
rhys.newbury@monash.edu

**Mustafa Mukadam**
Meta AI
mukadam@fb.com

**Dana Kulić**
Monash University
dana.kulic@monash.edu

**Akansel Cosgun**
Deakin University
akan.cosgun@deakin.edu.au

**Abstract:** We study gravitational pivoting, a constrained version of in-hand manipulation, where we aim to control the rotation of an object around the grip point of a parallel gripper. To achieve this, instead of controlling the gripper to avoid slip, we *embrace slip* to allow the object to rotate in-hand. We collect two real-world datasets, a static tracking dataset and a controller-in-the-loop dataset, both annotated with object angle and angular velocity labels. Both datasets contain force-based tactile information on ten different household objects. We train an LSTM model to predict the angular position and velocity of the held object from purely tactile data. We integrate this model with a controller that opens and closes the gripper allowing the object to rotate to desired relative angles. We conduct real-world experiments where the robot is tasked to achieve a relative target angle. We show that our approach outperforms a sliding-window based MLP in a zero-shot generalization setting with unseen objects. Furthermore, we show a 16.6% improvement in performance when the LSTM model is fine-tuned on a small set of data collected with both the LSTM model and the controller in-the-loop. Code and videos are available at https://rhys-newbury.github.io/projects/pivoting/

**Keywords:** Tactile Pose Sensing, In Hand Manipulation

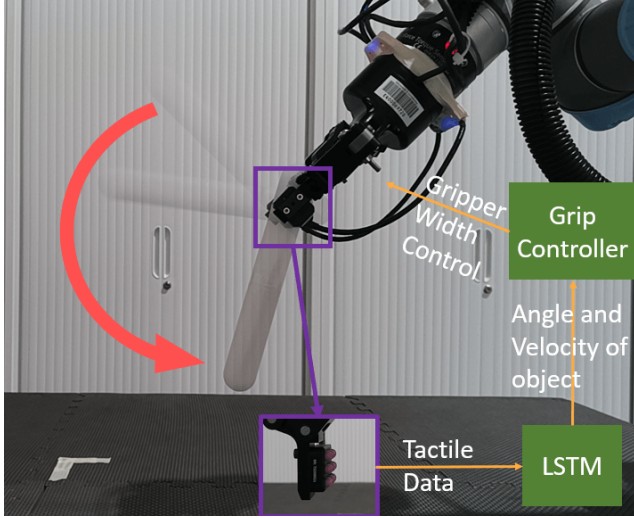

Figure 1: We study in-hand manipulation to rotate an object with gravitational pivoting. We design an LSTM model to predict the position and velocity of an object purely from tactile sensing. Our grip controller uses this prediction to modulate the width of the gripper and achieve a target angle.

6th Conference on Robot Learning (CoRL 2022), Auckland, New Zealand.

# 1   Introduction

The majority of past works in robotic manipulation either assume fixed grasps [1, 2] or aim to avoid any slipping during manipulation [3]. However, we focus on *embracing slip*, using object slip to increase the dexterity of simple grippers. This idea was explored by Chen et al. [4] where the grip on the object is loosened to allow an object to slip downwards with gravity. This paper explores a method of inducing rotation for in-hand manipulation using gravity. Such manipulation, known as pivoting, is key to performing tasks requiring an object to be at a specific relative angle to a gripper, such as stacking shelves [5].

This paper proposes a method for a robot with a parallel gripper to rotate a long object grasped away from its center of mass to a desired final relative orientation. This aims to allow parallel grippers to robustly reorient objects into a desired orientation without having to regrasp the object. To achieve this task, we focus on addressing two challenges: tracking the position of the object and controlling the gripper to allow for gravitational pivoting towards the target angle.

Vision-based methods to track the object often make use of an eye-in-hand camera. However, the gripper will often occlude the object, making it difficult to estimate the angle of the object accurately [6]. An alternative is to use an externally placed camera. However, this necessitates the robot moving to a fixed position in front of the camera for each manipulation. We use purely tactile information to track the object to avoid these issues. We design an LSTM-based neural network model, RSE-LSTM, which uses tactile information to predict a held objects' relative angular position and angular velocity.

Previous approaches to controlling the gripper often used model-based approaches, which required information about the object, such as shape, mass, and friction. In contrast, we design a simple gripper controller that assumes no a priori knowledge about the object parameters to allow for generalization to unseen objects.

We collect a real-world force-based tactile dataset, on ten household objects. This dataset is annotated with both angular position and velocity measurements. RSE-LSTM is trained on this dataset, and the results are reported with respect to both unseen data and unseen objects. We further validate our approach experimentally on unseen objects.

The contributions of our paper are threefold:

- An annotated dataset containing gravitational pivoting with 10 household objects.
- A LSTM-based neural network which can predict both the velocity and angle of an object using only tactile information.
- A grip controller, which can adjust the width of the gripper to allow an object to pivot in-hand to achieve a required relative angle.

# 2   Related Works

**Slip measurement.**  Slip detection is often framed as a binary problem, with machine learning models predicting either slip or no slip. This is achieved with the use of visual sensors [7], force-based sensors [8]. or optical sensors [9]. Various machine learning techniques have been used including: Support Vector Machines [10, 9], MLPs [11] and LSTM models [12]. Alternatively, Convolutional Neural Networks (CNN) have been used to both detect and classify the type of slip as either translational or rotational [13]. LSTM models have been used to determine the direction of rotational slip [14], or the overall direction of the combination of rotational and translational slip [15].

The domain of quantitative slip measurement is comparatively underexplored. Previous works measure the amount of translational slip using image-based tactile sensors [16]. Alternatively, visual gel-based tactile sensors have been used to measure the rotation angle using a model-based approach [17]. However, to our knowledge, the use of force-based tactile sensing has not been explored, which is the focus of this paper.

**Induced rotation.**  To induce rotation in a held object, previous work has made use of the external environment to apply a torque or force on the held object [18, 19, 20]. However, rotation can also be induced without any interaction with external objects. For example, the robot can perform a swinging motion using the end-effector, where the velocity of the swing aims to bring an object to a desired angle [21, 22, 23].

Alternatively, by loosening the grip on the object, gravity can be used to induce a rotation in the object [24, 25, 26, 27, 28, 29]. These approaches are model-based and rely on prior knowledge of important parameters of the system, such as shape, mass and friction of the held object. Our work assumes no prior knowledge about such parameters to allow for generalization to unseen objects.

**Induced translational slip.** Shi et al. [30] design a model-based approach to induce translational slip in an object by accelerating the gripper causing the object to slide in a desired way in-hand. Chen et al. [4] train a MLP to predict the velocity of an object which is undergoing translational slip. They feed the MLP the previous one second of observations to predict the sliding velocity of the object. A controller is then designed to achieve a target sliding velocity for the object. Our work extends this to the rotational case and makes use of an LSTM rather than providing a fixed length history. The allows the network to learn an encoding for the history, which may be more informative.

## 3 Problem Definition

Gravitational pivoting is a form of in-hand manipulation, where an object is rotated in hand by loosening the grip on the object. Consider a static gripper which holds an object away from its center-of-mass. If the gripper is closed tightly enough, the object should remain stationary inside the gripper. However, if the grip on the object is loosened slightly, gravity will induce a torque on the object, causing it to pivot inside the gripper.

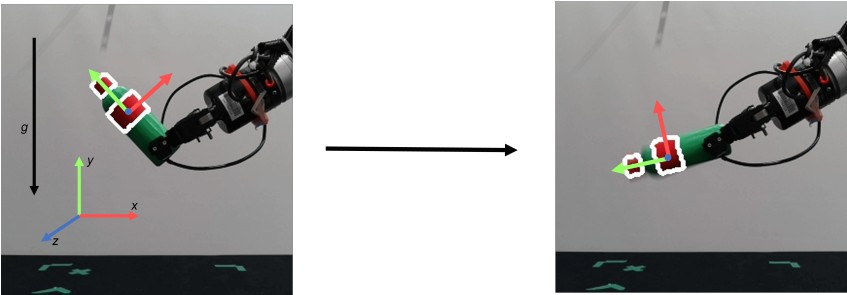

Figure 2: Gravitational Pivoting occurs when a parallel-jaw gripper loosens its grip on a object such that gravity induces a torque on the object. This torque will then cause a rotation to occur.

The coordinate system for this problem is defined in Figure 2. Gravity is defined to work in the negative $y$-direction in the global frame. We also define a rotating object-centric coordinate frame, where the $y$-axis is aligned with the long axis of the object.

Specifically, we consider pivoting tasks as follows. The robot starts with an object in hand, grasped away from the center-of-mass. The task is to rotate the object by a relative angle around the object-centric $z$-axis. We only consider achieving relative angles ($\alpha$) in the range $[0, 180]$ degrees, as we only consider allowing the object to fall and rotate. We assume that the robot has no a priori knowledge about the object and *only* has access to tactile information from the fingertips. This aims to allow generalizability by assuming no knowledge of the grasped objects. We constrain the type of objects to ones that have a prism-like shape, where one dimension, length, is much larger than the other two, width and depth. When these objects are held away from the center-of-mass, the torque induced on the object by gravity will be much larger than the downwards force at the contact points. Therefore, the objects are

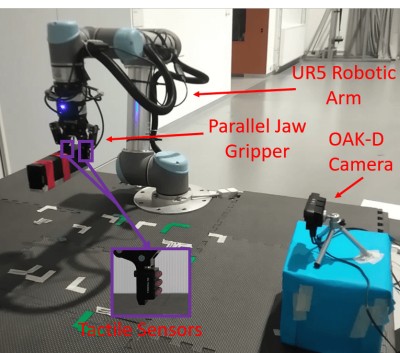

Figure 3: The hardware setup for both data collection and experiments.

more likely to undergo rotational slip when the grip is loosened, and the translational slip is assumed to be negligible. Prism-like objects include common household objects, such as bottles, boxes, and tools (such as hammers).

# 4 Data Collection

## 4.1 Hardware Setup

The robot hand consists of a Robotiq 2f-85 parallel-jaw gripper and a table-mounted UR5 robot arm. On each jaw of the gripper is a PapillArray tactile sensor by Contactile [31]. The parallel jaw gripper has a width of 85mm and can be controlled in 256 increments, for a resolution of 0.33mm per increment. The PapillArray sensors consist of 9 pillars, arranged in a 3 by 3 square, where each pillar provides a force and displacement measurement in each direction, and whether the pillar is in contact with the object. The sensor also provides global forces and torques in each direction, for a total of 142 measurements over the two sensors. In addition, an OAK-D camera is positioned to provide a side-on view of the robotic arm and held object, which will be used to record the ground truth angle (more details of the ground truth collection is provided in Section 4.3). The hardware setup for both data collection and experiments is shown in Figure 3.

## 4.2 Object set

We consider the pivoting task for a set of 10 different household objects. Similar to the object set used by Chen et al. [4], we consider two classes of objects: box-like objects and cylinder-like objects, with five objects in each class. The set of objects used are shown in Figure 4.

## 4.3 Methodology

A systematic methodology is used to collect a dataset of gravitational pivoting, outlined in the supplementary material. We use two different methodologies for controlling the gripper during rotation of the object:

- **Rotate To Stop**: The gripper is opened a fixed amount and the object is allowed to rotate until the object comes to a stationary position.
- **Angle Goal**: The controller (described in Section 5.2) is tasked to stop the object at an angle ($\alpha_{stop}$) from a set of angles ($A_{stop}$). The controller receives ground truth angle readings for the purposes of data collection.

To create a larger variation of friction properties, we collect data both with and without a layer of masking tape added to each object surface. In total there were 595 'Rotate To Stop'

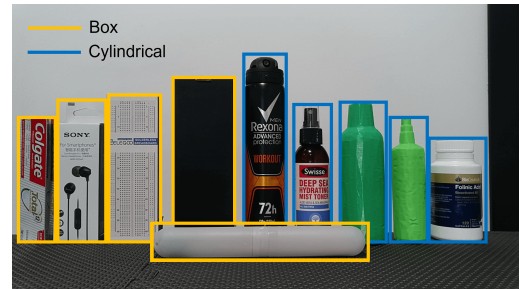

Figure 4: Objects used in this dataset. We distinguish between two classes of objects, box-like object and cylinder-like objects. We refer to the names of these objects throughout paper. The names are from left to right (back row): Toothpaste, Earbud, Breadboard, Magnet, Deodorant, Spray2, Shampoo, Spray1, Pill. The object at the front is labeled as Toothbrush.

and 971 'Angle Goal' sequences collected, after filtering invalid datapoints. The dataset is attached in the supplementary material.

## 4.4 Ground Truth Annotation

To measure the ground truth rotation of the grasped object, two distinctly colored blobs have been attached to each object. An external camera observes these blobs, and we then define the object-centric $y$-axis between the centroid of the two blobs. The orientation change between the initial object-centric $y$-axis and the current object-centric $y$-axis is used as the ground-truth angles. The position and angular velocity are filtered to ensure they follow a smooth signal. The details of the filter are in the supplementary material.

# 5 Proposed Approach

Our approach consists of two main components. A Rotational Slip Estimator LSTM (RSE-LSTM) and a Grip Controller. The system diagram is shown in Figure 1. From purely tactile information, the RSE-LSTM estimates the relative angle change between the initial and current object-centric $y$-axis. The Grip Controller uses the estimation and gravitational pivoting, aiming to reach a desired rotation relative to the initial angle.

## 5.1 Rotational Slip Estimator

The RSE-LSTM uses measurements from the tactile sensors and predicts both the current angular velocity ($\omega$) and relative angle of the object ($\alpha$) as the outputs. We found that calculating both $\alpha$ and $\omega$ improved the results of the model (Section 6.3).

The RSE-LSTM model consists of an LSTM, the outputs of which are passed to an MLP. Instead of using a sliding window (similar to [4]), using the hidden states of the LSTM could allow the model to use a longer history more effectively by learning a more feature-rich hidden state. The model runs at 60 Hz due to hardware limitations in training data collection.

## 5.2 Grip Controller

---
**Algorithm 1** Grip Controller

---
$goal \leftarrow \alpha_{stop}$
$error \leftarrow relative\_goal$
**while** $error > \epsilon_\alpha$ **do**
    $m \leftarrow$ Observation()
    $\alpha, \omega \leftarrow$ RSE-LSTM($m$)
    **if** $\omega < \omega_{min}$ and $t_{curr} - t_{prev} > t_{wait}$ **then**
        increase_gripper_opening()
        $t_{prev} \leftarrow t_{curr}$
    **end if**
    $\alpha_F \leftarrow$ fp($\alpha, \omega, d$)
    $error \leftarrow |\alpha_F - goal|$
**end while**
fully_close_gripper()

---

The grip controller algorithm is described in Alg.1. The grip controller uses the position and velocity estimated by the LSTM to achieve a target angle. While the object is not moving, the controller slightly opens the gripper. Once the object reaches the target angle, the gripper then closes. However, this alone constantly overshoots the target due to the delay in closing the gripper. To account for this, the velocity estimate is used to forward predict the angle (denoted by fp in Alg. 1), by a fixed delay ($d$) to stop the gripper early. The forward prediction equation is: $\alpha_F = \alpha + d \times \omega$.

## 6 Pose Estimate Model Evaluation

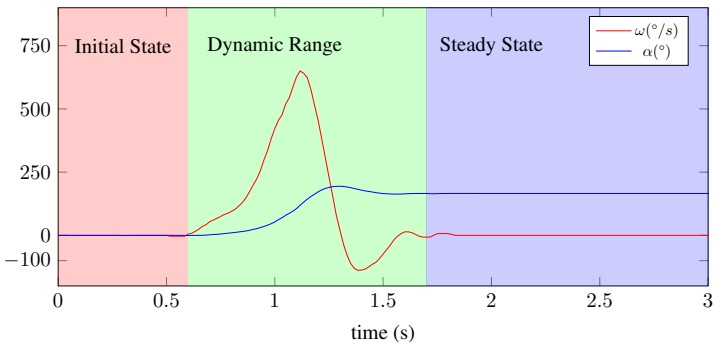

Figure 5: We split gravitational pivoting in to three sections, denoted by the three colors in the graph

## 6.1 Baseline

Chen et al. [4] found promising results using a sliding-window MLP in a similar task, where they predict the velocity of an object undergoing translational slip. Therefore, as a baseline, we consider the use of a sliding window **MLP**, which takes the last 15 steps of tactile data and predicts the angle and angular velocity at the next timestep. The parameters of this model were optimized using a sweep to maximize the performance on the test set, during random split experiments. In Section 9.10 we present results an ablation study on the length of the sliding window.

## 6.2 Experiments

We split the trajectory in to three sections (shown in Figure 5).

1. Initial State (IS): The object is held still, before any movement. The networks should be able to predict both 0 for the initial relative angle and 0 for velocity.
2. Dynamic Range (DR): The object is currently rotating. The models should track both position and velocity.
3. Steady State (SS): The object has finished rotating. The network should be able to predict the correct final angle and 0 velocity.

The dynamic range commonly lasts less than a second, thus necessitating learned estimation and control work in real-time. The objects also commonly reach velocities of up to 750°/s, making accurately predicting the velocities of the object difficult. The fast rotation of the object also increases the need for both the perception and control to be real-time, as any time delay in the system will magnify any errors.

We conduct three experiments, firstly experimenting on unseen data (Section 6.3), secondly experimenting on unseen objects (Section 6.4) and finally experimenting on unseen classes. The training and network parameters are detailed in the supplementary material with the unseen classes experiment results.

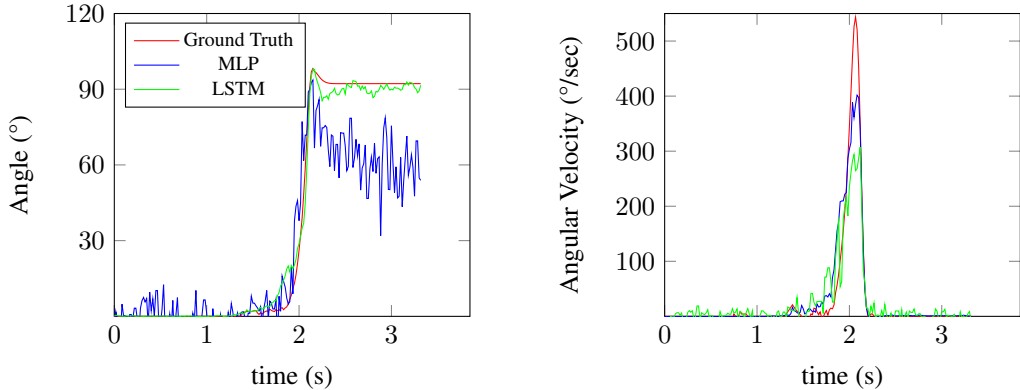

Figure 6: Example Tracking Performance. The LSTM produces a much smoother output compared to the MLP. The models display a consistent undershooting of the peak angular velocity.

## 6.3 Rotation Estimation

We train the previously described **LSTM** and **MLP** on the collected dataset. We varied the output of both models, to predict only $\alpha$, only $\omega$ or both $\omega$ and $\alpha$. If the model only outputs $\omega$, $\alpha$ will be recovered by integrating the signal. If the model calculates only $\alpha$, $\omega$ will be calculated by taking the derivative. An 80/20 train/test split of the data was used for training and the mean absolute error (MAE) is reported in Table 1.

| | Angular Error (°) | | | Angular Velocity Error (°/$s$) | | |
|---|---|---|---|---|---|---|
| | IS | DR | SS | IS | DR | SS |
| **LSTM Both** | 0.29±0.06 | **4.39±0.18** | **7.23±0.2** | **0.64±0.22** | 32.38±2.6 | 1.47±0.42 |
| **MLP Both** | 0.67±0.58 | 10.58±0.57 | 19.85±1.07 | 0.76±0.27 | 41.38±1.64 | 1.37±0.3 |
| **LSTM ($\omega$-only)** | **0.27±0.03** | 7.4±2.1 | 13.79±4.36 | 0.71±0.20 | **31.97±2.36** | **0.81±0.26** |
| **MLP ($\omega$-only)** | 0.39±0.11 | 12±1.14 | 22.21±2.35 | 1.38±0.5 | 36.93±2.01 | 1.92±0.57 |
| **LSTM ($\alpha$-only)** | 0.31±0.03 | 4.52±0.21 | 7.23±0.26 | 1.74±0.32 | 40.13±1.86 | 4.8±0.25 |
| **MLP ($\alpha$-only)** | 1.48±0.55 | 10.41±0.67 | 19.79±1.11 | 3.52±1.28 | 65.07±1.62 | 8.31±0.81 |

Table 1: Testing results on unseen data. Each network is trained 5 times. IS, DR, SS represent the three sections of motion, Initial State, Dynamic Range and Steady State, respectively. $\omega$ represents the angular velocity, and $\alpha$ represents the angular position.

**LSTM Both** out-performed the other models in terms of the position error, slightly outperforming **LSTM (α-only)** in terms of the position error in all three stages. **LSTM Both** also performed similarly to **LSTM ω-only** in terms of the velocity error. The MLP performance was lower than the LSTM for all stages for both velocity and position error.

An additional benefit of the LSTM over the MLP is that the LSTM produced much smoother results than the MLP. This can be seen in Figure 6, where the MLP predictions are more noisy compared to LSTM. This is problematic when paired with the Grip Controller, making it challenging for the controller to achieve precise angles and also more prone to early failure.

### 6.4 Unseen Objects

For the next experiments, we only consider models that output both $\alpha$ and $\omega$, as they were the best performing models. We investigate the performance of the models when generalized to unseen objects. We trained the LSTM and MLP on 9 objects and tested on the remaining 1 unseen object. The performance of the models decreased compared to training on previously seen objects. This is most likely due to the model struggling to generalize to objects that have unique properties compared to the other objects in the training set. An example is the 'Toothpaste' object, where the position error is much larger for both the LSTM and MLP, as shown in Figure 7.

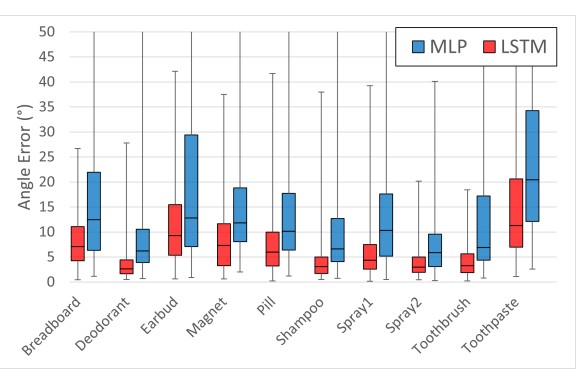

Figure 7: Distribution of position errors for each unseen object.

## 7 Controller Evaluation

### 7.1 Baselines

We consider two baselines:

- **Oracle**: Ground truth angle and angular velocity measurements for the Grip Controller are taken from visual data. This provides an upper bound on the performance of the gravitational pivoting system.
- **MLP**: The MLP described in Section 6.1

### 7.2 Experimental Setup

The experimental methodology followed was same as the 'Angle Goal' data collection process described in Section 4.

We experimentally tested the performance on unseen objects using the models from Section 6.4. During experiments both the ground truth and predicted $\alpha/\omega$ were recorded. These measurements were used to compute the MAE of the model within the 3 sections as defined in Section 6.2 (IS, DR and SS).

The most important error in the controller experiments is the error in the final angle achieved by the controller, denoted as **target error (TE)**. We also report the **failure rate (FR)** which is the percentage of trials for which the controller either dropped the object or became stuck, predicting an $\omega > \omega_{min}$ while the object is stationary.

To address the model generalization to the real-world setup, we further collected a small amount of 'Angle Goal' data with the controller receiving angle measurements from the trained RSE-LSTM model instead of the Oracle to test the effects of fine-tuning the controller on the desired scenario. The data collection parameters are detailed in the supplementary material.

## 7.3 Results

The results of the real-world experiments are shown in Table 9. While the LSTM outperformed the MLP for all metrics, the tracking error was much larger than the error seen on the test set. This is likely because the training dataset does not represent all scenarios seen in real-world testing such as overshooting the target angle and the gripper not closing.

Furthermore, the target error was much larger than the *Oracle* baselines. This can be attributed to the previously stated tracking inaccuracies, particularly in dynamic range, resulting in the model predicting the target angle was reached either too late or too early.

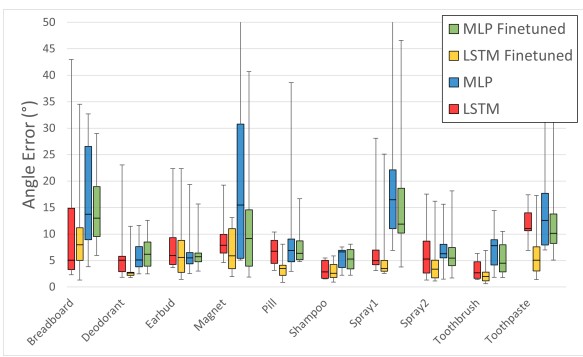

Figure 8: The target-error for all unseen object experiments. The fine-tuned LSTM outperformed all other methods for 8 out of the 10 objects.

As shown in Figure 8, fine-tuning improved the results on most objects. The distribution of the box plot follows a similar pattern to Figure 7. This suggests some objects are harder for our approach to generalize to. This could be further improved by collecting data on a wider range of objects.

|  | Failure Rate (%) | Target Error (°) |
|---|---|---|
| **Oracle** | 0 | 3.95±3.02 |
| **LSTM** | **3.3** | 15.2±4.69 |
| **MLP** | 6.7 | 23.08±8.31 |
| **LSTM-finetuned** | 4.44 | **12.67±5.31** |
| **MLP-finetuned** | 10.00 | 23.10±8.17 |

Table 2: Results of fine-tuned models with experimental setup. The LSTM-finetuned outperforms the other models in terms of the target error

## 8 Limitations and Future Work

One of the main challenges of gravitational pivoting is that to avoid extra motion of the end-effector, the object can only rotate in one direction. Hence, it is hard to recover from errors when the object moves past the target angle. To minimize the effect of the issue, we aim to rotate the object slowly, by slowly loosening the grip until some velocity is observed. However, the object can still rotate in the magnitude of 500°/s. The fast speeds necessitate accurate stopping conditions, which makes accurate target reaching very difficult. Furthermore, there is a large delay in commanding the gripper to close and the object coming to a rest, at around 0.83s. The stopping condition uses forward prediction proportional to both the delay and velocity, therefore, any errors in velocity prediction are magnified by a large time delay. A potential solution for both of these issues would be to slow the object down as it is pivoting. This could be achieved using a learning-based controller, which could make smarter decisions on when to speed up or slow down pivoting. Another potential solution would be to increase the speed of the data-collection process and in turn allow RSE-LSTM to operate at a higher frequency. This would allow more reactive control to fast changes in position and velocity. Another limitation of our work is that the performance degrades when working with unknown objects. This could be improved with collecting more varied data on a wider range of objects. Further investigation is needed at looking at the effect of out of distribution testing objects, for examples lighter objects will likely be harder to rotate. Our work aims to show that tactile information is sufficient for this problem. While it does show promising results, further work needs be done to improve the results, to make both the estimation and the controller more robust.

**Acknowledgments**

We would like to thank Lily Tran for help editing the manuscript.

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
