# OpenReview forum: "In-Hand Gravitational Pivoting Using Tactile Sensing"
_robot-learning.org/CoRL/2022/Conference — CoRL 2022 Poster_

### Official Review · Reviewer_Yu3a · 2022-07-28

**Originality:** Good
**Technical Quality:** Very Good
**Clarity Of Presentation:** Very Good
**Impact:** 3

**Recommendation:**

Weak Accept: I recommend accepting the paper, but will not argue for my recommendation if the majority of other reviewers have a different opinion.

**Summary:**

This paper looks at the in-hand manipulation of objects with a two fingered gripper by relaxing the grip to allow the object to rotate under gravity. A simple 2 state controller is used (slightly open / hold) depending on the object's angle and rate of change of angle, aiming to reach a desired angle of rotation. These are estimated using a MLP and LSTM on tactile data from two PapillArray sensors mounted on a Robotiq parallel-haw gripper. The networks are trained on several objects and tested on novel objects. Overall, the LSTM works best and is effective with an angle error of 5-10deg.

**Issues:**

Please see comments above.

**Quality Of The Limitations Section:**

Limitations are addressed clearly

**Reviewer Expertise:**

3: The reviewer is fairly confident that the evaluation is correct

**Robotics Focus:**

Sufficient demonstration on hardware

**Strengths And Weaknesses:**

Strengths
- Fairly novel task for tactile sensing. Could be used as a benchmark to compare sensors, grippers etc.
- Nice combination of machine learning and control.
- I learnt that you also need to use the rate of change of angle when trying to reach a desired angle, as a correction on the latency in the system.

Weaknesses
- The controller has just two preset states (close or slightly open). How is that tuned to the different object sizes? Is that done manually? As you have a tactile sensor it would have been nice to use the sensor to set how closed and open you need to be.
- Prediction of object angle is a basic capability that there are many ways of doing with different sensors (e.g. using image moments etc). What made your particular sensor well suited for this? I had no sense of how the tactile data was interpreted by the ML method to make the prediction of angle.
- The scope seemed limited. Once the method was shown to work, it would have been good to see some other demonstration of its capability or more explanation of aspects of it to broaden the scope. E.g. rather than gravity you could use translate the gripper so the object is turned by knocking against something.
- You could have been more complete in your referencing - e.g. there are plenty more on slip and related topics (including a review article by the inventors of the sensor in this paper). Not a major weakness, but I don't think there is any limit on referencing at this conference.
- How suitable is the sensor for this task. As I understand it, it has a relatively low dimensional output (3 x 3 array) that is well suited for slip detection. However, you are using it for object pose estimation, where I would have thought more detailed sensors e.g. the DIGIT would be more effective. I would have liked to have seen some benefits of the tactile sensor you are using coming through in this study.

**Summary Of Recommendation:**

I like the paper, but veer towards a weak accept because the scope is narrow and the problem is somewhat straightforward to solve. I also did not learn much about how the tactile sensor data is intererpreted to allow the task to be succesful, which would have been interesting.

---

> ### Author Response · Authors · 2022-08-23
> **Author Response to Reviewer Yu3a (Part 1 of 2)**
>
> Firstly, we would like to thank you for your detailed comments and feedback. We have tried our best to update the manuscript according to your suggestions. The updated manuscript can be found at: https://openreview.net/pdf?id=sCyKrS0k5a9 (login required)
>
> >The controller has just two preset states (close or slightly open). How is that tuned to the different object sizes? Is that done manually? As you have a tactile sensor it would have been nice to use the sensor to set how closed and open you need to be.
>
> The gripper width for the ‘slightly open’ state for the ‘Rotate to Stop’ data collection and the ‘closed’ state were manually determined. However the controller itself gradually opens the gripper from the ‘closed’ state so only the ‘closed’ state is required to be known. We chose to manually measure these values as we felt that measuring a good ‘close’ width with the tactile sensor was not directly related to the project, although it could be integrated later to make the system more automated and generalisable.
>
> >Prediction of object angle is a basic capability that there are many ways of doing with different sensors (e.g. using image moments etc). What made your particular sensor well suited for this? I had no sense of how the tactile data was interpreted by the ML method to make the prediction of angle.
>
> We chose to focus on tactile sensors, as our research question was focused on if tactile sensors could be used for this sort of task. This could also be achieved with other sensors, such as cameras. However, there are various advantages of tactile sensors for this task. Tactile sensors are mounted on the robot, so a separately mounted system for sensing is not required. A wrist mounted camera would suffer from many occlusions, and this presents its own set of challenges.
>
> As the LSTM is a deep learning method, we treat it as a black box. Interpretability of how the model estimates angle would be an interesting direction for future work but due to the complexity of such a task we chose to not pursue that direction.
>
> >The scope seemed limited. Once the method was shown to work, it would have been good to see some other demonstration of its capability or more explanation of aspects of it to broaden the scope. E.g. rather than gravity you could use translate the gripper so the object is turned by knocking against something.
>
> We acknowledge that this could be used for other applications, but we only focused on only the rotation task. Other previous works have focused on using the environment to rotate the grasp, but we believe the use of gravity for pivoting is an interesting direction of work and very useful for robotic applications. Previous works have also focused on the related task of translational slip due to gravity, but not rotation so we focused on that. but that a similar approach for both could be taken.

---

> > ### Author Response · Authors · 2022-08-23
> > **Author Response to Reviewer Yu3a (Part 2 of 2)**
> >
> > >You could have been more complete in your referencing - e.g. there are plenty more on slip and related topics (including a review article by the inventors of the sensor in this paper). Not a major weakness, but I don't think there is any limit on referencing at this conference.
> >
> > There is plenty of literature on this topic, and we believe it is not feasible to cite every paper. We have tried to cover the most highly related papers in our literature review and we have added references to the survey you mentioned as well as other papers as pointed out by the reviewers. If there are specific references you think it would be appropriate for us to cite, please let us know.
> >
> > >How suitable is the sensor for this task. As I understand it, it has a relatively low dimensional output (3 x 3 array) that is well suited for slip detection. However, you are using it for object pose estimation, where I would have thought more detailed sensors e.g. the DIGIT would be more effective. I would have liked to have seen some benefits of the tactile sensor you are using coming through in this study.
> >
> > We have provided extra detail on the sensor, which provides 142 outputs. We believe the chosen sensor is very suitable for this task, as the changes in force on the gripper can be very small, due to the loose grip on this object. We performed early experimentation using the DIGIT sensor, and found that with the default gel, the loose grip on the object would not cause a very deep imprint and, hence, very little information. This was not a rigorous study, therefore we will not add the details into the paper. Furthermore, by changing the gel to be more malleable and better able to detect tactile readings from a loose grasp, this sensor then could be used for this task.
> >
> > However, we found the PapilArray sensors quite suitable due the ability for it to detect small changes in force readings while still providing a lot of information to the model. One area where the sensor may not be suitable is for rotating small objects, such as a kitchen utensil. For such objects the gaps between the pillars become relevant and the object held no longer smoothly rotates as it does for larger objects.

---

### Official Review · Reviewer_ynxK · 2022-07-29

**Originality:** Good
**Technical Quality:** Very Good
**Clarity Of Presentation:** Excellent
**Impact:** 3

**Recommendation:**

Weak Accept: I recommend accepting the paper, but will not argue for my recommendation if the majority of other reviewers have a different opinion.

**Summary:**

This paper leverages gravity to achieve in-hand manipulation of prism-like objects via pivoting objects to a desired angle. The proposed solution has two components: (1) a Rotational Slip Estimator LSTM (RSE-LSTM) that estimates the relative angle change and velocity of the object based on tactile information and (2) a grip controller that uses RSE-LSTM to open the gripper until the target angle for the object is achieved. Since the control is based on the feedback of a learned tactile module, the system assumes no knowledge of object parameters, other than the fact that the object is prism-like and will thus experience minimal translational slip relative to the rotational motion. For the RSE, an LSTM-based model outperforms an MLP-based model on unseen objects. It is also shown that it improves performance to predict angle and velocity. The LSTM model leads to better controller performance, (compared to MLP), although even with fine-tuning both lagged behind the oracle's performance.

**Issues:**

As stated above it would be great to have a bit more discussion (perhaps in the appendix, since the paper is space constrained) about how to improve the generalization to new objects (or what is a limiting factor preventing this). This could include discussion of what properties of objects make generalization difficult. For example, there is a comment in Section 6.4 that error for the toothpaste is higher. What about the toothpaste causes this?

In collecting the dataset, there were two methodologies for controlling the gripper, "Rotate to Stop" and "Angle Goal". Was there a significant difference between the data collected by these two methodologies? Would there be an impact in using data gathered by one over the other? Additionally, I believe the "Angle Goal" controller leverages the RSE-LSTM, which is trained based on data collected by this process. It would be helpful to clarify this.

In algorithm 1, is there any mechanism that prevents the gripper from opening past the point where the object would fall out? Would it be possible to integrate such a mechanism?

If possible, videos of pivoting the various objects (particularly the unseen objects) would strengthen the paper.

These two related papers might be worth discussing and citing:
- Sintov, Avishai, and Amir Shapiro. "Swing-up regrasping algorithm using energy control." In ICRA, pp. 4888-4893. IEEE, 2016.
- Holladay, Anne, Robert Paolini, and Matthew T. Mason. "A general framework for open-loop pivoting." In ICRA, pp. 3675-3681. IEEE, 2015.

**Quality Of The Limitations Section:**

Limitations are addressed clearly

**Reviewer Expertise:**

4: The reviewer is confident but not absolutely certain that the evaluation is correct

**Robotics Focus:**

Sufficient demonstration on hardware

**Strengths And Weaknesses:**

This paper prevents a novel method for pivoting an object in-hand in order to modify the object's orientation. In contrast to many other in-hand manipulation systems, the system does not assume any knowledge of object properties, which should lead to increased generalization. The issue of accurate and very dynamic manipulation, as tackled in this paper, is a tricky one. As stated in the limitations, the relatively simple controller serves as a good jumping off point for future work.

The paper is well-written, with the structure clearly laying out each contribution. Section 3 was particularly helpful in clearly stating the problem and the guiding assumptions. The experimental evaluation section was comparatively extensive. Given that the closest related work used an MLP in the context of translational slip, it made sense to compare LSTM and MLP.

As stated above, accurate and dynamic manipulation is tricky. However, it would strengthen the paper to better understand what could be done to improve the generalization to new objects (particularly since generalization is a strength of the approach). Is this primarily a weakness of the estimator or the controller? The paper mentions that some objects are harder to generalize to. What are the properties of these objects that contribute to this?

It would also be great to include some videos of the system in action as part of the supplementary material.

**Summary Of Recommendation:**

I am weakly recommending the acceptance of this paper. The paper presents an interesting in-hand manipulation method that leverages tactile sensing to perform dynamic pivoting. The formulation does not rely on any object properties (aside from being prism-shaped) and thus can generalize to many objects, although there may be some more tweaks or training needed to fully fulfill this promise. The paper is well-organized and provides appropriate in-hand manipulation demonstrations.

---

> ### Author Response · Authors · 2022-08-23
> **Author Response to Reviewer ynxK (Part 1 of 2)**
>
> Firstly, we would like to thank you for your detailed comments and feedback. We have tried our best to update the manuscript according to your suggestions. The updated manuscript can be found at: https://openreview.net/pdf?id=sCyKrS0k5a9 (login required)
>
> >As stated above, accurate and dynamic manipulation is tricky. However, it would strengthen the paper to better understand what could be done to improve the generalization to new objects (particularly since generalization is a strength of the approach). Is this primarily a weakness of the estimator or the controller? The paper mentions that some objects are harder to generalize to. What are the properties of these objects that contribute to this?
>
> >As stated above it would be great to have a bit more discussion (perhaps in the appendix, since the paper is space constrained) about how to improve the generalization to new objects (or what is a limiting factor preventing this). This could include discussion of what properties of objects make generalization difficult. For example, there is a comment in Section 6.4 that error for the toothpaste is higher. What about the toothpaste causes this?
>
> Generalisation to new objects is mainly limited currently by the performance of the estimator from our experimentation. This is because the estimator error is the main contributing factor to the higher target error seen when evaluating the controller, as evidenced by the low target error produced by the oracle system.
>
> To improve the performance of the estimator we would primarily focus on increasing the range of objects the model was trained on, exposing it to a broader range of shapes, weights and surface properties than in our study. This can be seen in the unseen class experiment in Section 9.4 which shows that the estimator struggles to generalise to objects with properties very different from the training set. This is also a likely reason for the underperformance of the toothpaste, which was the only object we used that underwent inelastic deformation throughout the period of experimentation. However, to make more conclusive statements on the exact properties that make objects harder to generalise to, we would recommend further experimentation on a broader object set.
>
> >It would also be great to include some videos of the system in action as part of the supplementary material.
>
> Thank you for this suggestion. We are currently developing a video and will be attaching it to this submission in the coming days. We have tried to provide prompt answers to all other questions, but the video will take longer to finalise. We appreciate your patience.

---

> > ### Author Response · Authors · 2022-08-23
> > **Author Response to Reviewer ynxK (Part 2 of 2)**
> >
> > >In collecting the dataset, there were two methodologies for controlling the gripper, "Rotate to Stop" and "Angle Goal". Was there a significant difference between the data collected by these two methodologies? Would there be an impact in using data gathered by one over the other? Additionally, I believe the "Angle Goal" controller leverages the RSE-LSTM, which is trained based on data collected by this process. It would be helpful to clarify this.
> >
> > The data collected by both data methods did have significant differences. The “Rotate to Stop” method opened the gripper to a fixed ‘loose’ grip width that would allow for rotation without dropping the object. The object would then rotate until it came to rest when its centre of gravity was directly below the grasp point. As the gripper was not closed to obtain a firm grasp at the end of this process, the RSE-LSTM could not learn how to handle extra forces from the gripper closing that were present in the controller experiments, and so when the grasp tightened the RSE-LSTM always predicted an angle of 0 degrees.
> >
> > The “Angle Goal” method instead used our control strategy with ground truth angle measurements obtained from  the Oracle system. This allowed it to gain experience with motion resulting from our control strategy. As this data collection always ended in the grasp tightening on the object, the RSE-LSTM also learnt to hold its current prediction constant when the forces from the tactile sensor increased due to the gripper closing.
> >
> > To clarify, the “Angle Goal” data collection does not use the RSE-LSTM, it uses the oracle. However, our fine tuning data does use the RSE-LSTM to allow the model to train on the exact motion seen in the experimentation with the controller. We have added information into Section 4.3 and Section 7.2 to clarify this point.
> >
> > In terms of performance gain from both data collection methods, the “Angle Goal” methodology should be better for controlling the gripper using our control strategy, while the “Rotate to Stop” data would be better for the tracking task of objects falling under gravity and stopping naturally.
> >
> > >In algorithm 1, is there any mechanism that prevents the gripper from opening past the point where the object would fall out? Would it be possible to integrate such a mechanism?
> >
> > There does not currently exist a method to stop the object falling out, the way our method aims to achieve this is by minimising the amount of gripper movement, such that the pressure is slowly released, with the aim being that the object will rotate before falling out of the gripper.
> >
> > >These two related papers might be worth discussing and citing:
> > Sintov, Avishai, and Amir Shapiro. "Swing-up regrasping algorithm using energy control." In ICRA, pp. 4888-4893. IEEE, 2016.
> > Holladay, Anne, Robert Paolini, and Matthew T. Mason. "A general framework for open-loop pivoting." In ICRA, pp. 3675-3681. IEEE, 2015.
> >
> > Thank you for pointing us to these interesting papers, we have updated our related work to include them.

---

### Official Review · Reviewer_Wnm1 · 2022-07-30

**Originality:** Good
**Technical Quality:** Good
**Clarity Of Presentation:** Good
**Impact:** 3

**Recommendation:**

Weak Accept: I recommend accepting the paper, but will not argue for my recommendation if the majority of other reviewers have a different opinion.

**Summary:**

This work studied pivoting objects using gravity with tactile sensors. It proposed a learning-based method to track the angle and velocity of the object in hand using force-based tactile sensors. The tracking model is used in a closed-loop control to pivot the object to a target angle. The proposed LSTM model outperforms the baseline MLP models in the tracking and closed-loop control tasks. In addition, the experiments show promises of transferring to unseen objects.

**Issues:**

As discussed in Weaknesses.

**Quality Of The Limitations Section:**

Limitations are addressed clearly

**Reviewer Expertise:**

4: The reviewer is confident but not absolutely certain that the evaluation is correct

**Robotics Focus:**

Sufficient demonstration on hardware

**Strengths And Weaknesses:**

Strengths:
- It is an important topic of in-hand manipulation that can benefit from tactile sensors.
- The proposed models achieve promising results and outperform baselines in perception and control tasks.
- The held-out experiments also show promises to generalization to unseen objects.
The paper is well written and easy to follow.

Weaknesses:
- The comparison between MLP and LSTM seems unfair. The MLP only has the last 15 timestamps, but LSTM has access to longer sequences. It is unclear whether more data or the better model causes the improvement. It will be clearer if there are experiments when the LSTM model also uses the last 15 timestamps.
- As the author mentioned in the Limitation section, the controller is relatively simple. The model requires relatively large movements to observe, which will introduce rather large errors. The author could consider comparing different controllers to achieve better performances.
- The physical properties of the objects should be described, like sizes, mass, center of mass, and surface roughness. This helps to interpret the data distribution, and see the generalization performance to different physical properties.
- The author should also discuss the range of suitable objects for this method. Light objects might be challenging to pivot based only on gravity.
- The work can be improved with video demonstrations to show tactile signals, real-time tracking, and the control policy.


**Summary Of Recommendation:**

This work provided a learning-based method for gravitational pivoting with real-time tactile feedback. The proposed models showed promising results and outperformed the baseline model. The perception result was further evaluated with a controller for application. While overall is good, it could further be improved by exploring more advanced controllers, models, fair comparison between baselines, and more details about the generalization.

---

> ### Author Response · Authors · 2022-08-23
> **Author Response to Reviewer Wnm1**
>
> Firstly, we would like to thank you for your detailed comments and feedback. We have tried our best to update the manuscript according to your suggestions. The updated manuscript can be found at: https://openreview.net/pdf?id=sCyKrS0k5a9 (login required)
>
> >The comparison between MLP and LSTM seems unfair. The MLP only has the last 15 timestamps, but LSTM has access to longer sequences. It is unclear whether more data or the better model causes the improvement. It will be clearer if there are experiments when the LSTM model also uses the last 15 timestamps.
>
> One advantage of recurrent based models, such as a RNN, GRU or LSTM is that they are able to work on arbitrary length historys. Contrasting to this, MLP will have to work on a fixed window size. We have added details in Section 9.10 on an ablation study we have performed to find this window size. Furthermore, we have also added details in Section 9.11 on an ablation study we have performed where an RNN and a GRU are trained to track the angle of unseen objects.
>
> >As the author mentioned in the Limitation section, the controller is relatively simple. The model requires relatively large movements to observe, which will introduce rather large errors. The author could consider comparing different controllers to achieve better performances.
>
> We agree with your comment that the controller is relatively simple, this is because the focus of our paper is addressing the sensing complexity and showing that even a simple controller can perform well with an accurate angle prediction. We chose not to compare different controllers as the control algorithm is not the focus of our work.
>
> One limitation of our work is that the delay of 0.83s for the gripper to react to control commands, makes precise control impractical. For future work, we believe a more reactive gripper would be needed in conjunction with a more complex controller.  We strongly believe future work in this area should design more complex controllers, which can better modulate the speed of the object, e.g., slowing the object down if the object is moving too fast. Furthermore, we believe an interesting direction for this would be a learning-based controller, which can dynamically adjust the gripper. We have refined the discussion on this in the Limitations section.
>
> >The physical properties of the objects should be described, like sizes, mass, center of mass, and surface roughness. This helps to interpret the data distribution, and see the generalization performance to different physical properties.
>
> Thank you for this useful suggestion. We have added Section 9.12 (in the supplementary materials), detailing object properties. We do not have equipment to measure the surface roughness but we have detailed important physical properties of each object.
>
> >The author should also discuss the range of suitable objects for this method. Light objects might be challenging to pivot based only on gravity.
>
> Thank you for this suggestion, we have added discussion on this point into the limitations section. The weight of the object did not seem to have a large effect on the series of objects we used, however, this may change once a larger change in weight is tested, or if the testing object is outside of the training distribution.
>
> >The work can be improved with video demonstrations to show tactile signals, real-time tracking, and the control policy.
>
> Thank you for this suggestion. We are currently developing a video and will be attaching it to this submission in the coming days. We have tried to provide prompt answers to all other questions, but the video will take longer to finalise. We appreciate your patience.

---

### Official Review · Reviewer_R21W · 2022-07-31

**Originality:** Good
**Technical Quality:** Very Good
**Clarity Of Presentation:** Good
**Impact:** 3

**Recommendation:**

Strong Accept: I recommend accepting the paper and will argue for my recommendation even if other reviewers hold a different opinion.

**Summary:**

The paper presents an approach for controlled pivoting of objects using tactile sensing. The outcomes of an LSTM approach are compared to an MLP based approach and shown to be superior. Vision is used as a ground truth.

**Issues:**

* In the first sentence of the introduction - it seems that such general claims on robotic manipulation would be better served by citing review papers rather than these specific pieces of work.

* It seems that the following paper from Bristol would be a relevant addition to the literature review - 'Slip Detection With a Biomimetic Tactile Sensor' 2018.

* The motivation for adding masking tape to the objects was not explained - we are simply told that friction is augmented. I assume this is to make the surface friction consistant across the objects (which simplifies the pivoting problem somewhat). Could the authors also make it clear that this was applied to all trials. I first read this as the tape being a variable.

* Why are two of the objects covered in green paper but the others are not?

* Are the objects full or empty? For example the deoderant and pills.

* We are not told how much control is possible over the gripper, in terms of grasp aperture resolution. It would be interesting to know what is the smallest amount you can reliable move one of these grippers and how much your controller has to move it.

* How are the 2x9 papillae of the tactile sensors combined to get a force output? Or are all 18 ouputs fed into the models?

*  I could not make sense of Table 1 for sometime until I read in Section 7.2 that the headings were defined in section 6.2. Though I had read section 6.2, I did not recall the described labels when seeing the acroynm, as there was quite a bit of space between 6.2 and the table. There is no harm in reminding the reader in the caption of the table and using the same acroynms elsewhere, such as on Figure 5.

* The benefits of slower rotational motion is argued in Section 8, but a counter-argument would be that this would slow down any processes that the robot is engaged in. An example is given earlier in the paper of shelf stacking, which would be a repetitive industrial task where speed is prized.

* I think a current limitation of the method is the reliance on the tape to normalise the frictional properties of the objects and would like to see this discussed in the limitations section.

**Quality Of The Limitations Section:**

Limitations are addressed clearly

**Reviewer Expertise:**

4: The reviewer is confident but not absolutely certain that the evaluation is correct

**Robotics Focus:**

Sufficient demonstration on hardware

**Strengths And Weaknesses:**

Strengths

The problem of controlled gravity based pivoting has been addressed by several other papers recently and so is a current topic of interest in robotics research..

Weaknesses

The MLP approach that the system is compared against is not well described and motivated in the main body. Though the parameters of the MLP are given in the supplemental material, it is not clear how these were determined and why the approach of [5] was selected as inspiration. Given the key antagonistic role the MLP approach plays in this work, the authors must convince the reader that they didn't intentionally design / tune a system to have poor performance.

The paper is unclear at times and it took me some backtracking to make sense of the results.

**Summary Of Recommendation:**

The paper is generally interesting and the arguments for the reason for investigating the problem are compelling. My major qualm with the paper comes from the MLP comparison method, which, as mentioned in the weaknesses, I think needs more text to justify why it was selected (compared to the approaches of oether papers) and how it was tuned. Until this is done, the outperformance claim is somewhat weak.

I have listed some minor issues in the following section which affect paper clarity. I mention this here as there were quite a few of these, which contributed to a sense of frustration by the time I finished reading the manuscript.

---

> ### Author Response · Authors · 2022-08-23
> **Author Response to Reviewer R21W (Part 1 of 2)**
>
> Firstly, we would like to thank you for your detailed comments and feedback. We have tried our best to update the manuscript according to your suggestions. The updated manuscript can be found at: https://openreview.net/pdf?id=sCyKrS0k5a9 (login required)
>
> >The MLP approach that the system is compared against is not well described and motivated in the main body. Though the parameters of the MLP are given in the supplemental material, it is not clear how these were determined and why the approach of [5] was selected as inspiration. Given the key antagonistic role the MLP approach plays in this work, the authors must convince the reader that they didn't intentionally design / tune a system to have poor performance.
>
> We chose to use the work of [5] as an inspiration, as they study a very similar problem to our work, however, looking at the translational slip case. This work was also very recently published, which made the comparison more suitable. The hyperparameters of the MLP were determined using a sweep, and we used the best model. We have expanded this detail in Section 9.5 of the Supplementary Material. Details have also been added into Section 6.1 in the main paper. We have also added Section 9.10 which details an ablation study performed on the size of the running window.
>
> >In the first sentence of the introduction - it seems that such general claims on robotic manipulation would be better served by citing review papers rather than these specific pieces of work.
>
> Thank you for your suggestion, we have updated this sentence to cite appropriate reviews.
>
> >It seems that the following paper from Bristol would be a relevant addition to the literature review - 'Slip Detection With a Biomimetic Tactile Sensor' 2018.
>
> Thank you for pointing out this interesting paper. This paper has been added into our Related Work section.
>
> >The motivation for adding masking tape to the objects was not explained - we are simply told that friction is augmented. I assume this is to make the surface friction consistant across the objects (which simplifies the pivoting problem somewhat). Could the authors also make it clear that this was applied to all trials. I first read this as the tape being a variable.
>
> The masking tape on the surface was a variable. We collected data for each object with and without tape. We chose to continue using tape to create a larger variety of friction properties for each object. We have refined the text to make sure this was more clear.
>
> >Why are two of the objects covered in green paper but the others are not? Are the objects full or empty? For example the deoderant and pills.
>
> Thank you for your comment. Two of the objects are 3D printed from previous work completed in the lab. We have added a new Section in the Supplementary Material (Section 9.9), which details the property of each object, including whether it is full or not.

---

> > ### Author Response · Authors · 2022-08-23
> > **Author Response to Reviewer R21W (Part 2 of 2)**
> >
> > >We are not told how much control is possible over the gripper, in terms of grasp aperture resolution. It would be interesting to know what is the smallest amount you can reliable move one of these grippers and how much your controller has to move it.
> >
> > We have 256 increments to move the 85mm gripper resulting in a resolution of approximately 0.33mm. This has been added in to the text in Section 4.1
> >
> > >How are the 2x9 papillae of the tactile sensors combined to get a force output? Or are all 18 ouputs fed into the models?
> >
> > For each pillar the sensor provides displacement measures, force measures and a flag indicating whether the pillar is in contact with an object. Furthermore, the sensor also provides global forces and global torques, for a total of 142 outputs. We concatenate these together into a vector before feeding it into the LSTM/MLP. We have updated the text in Section 4.1 to make this more clear.
> >
> > >I could not make sense of Table 1 for sometime until I read in Section 7.2 that the headings were defined in section 6.2. Though I had read section 6.2, I did not recall the described labels when seeing the acroynm, as there was quite a bit of space between 6.2 and the table. There is no harm in reminding the reader in the caption of the table and using the same acroynms elsewhere, such as on Figure 5.
> >
> > Thank you for this helpful suggestion. We have updated the caption to include and define the acronym in the tables, both for Table 1 and later Tables.
> >
> > >The benefits of slower rotational motion is argued in Section 8, but a counter-argument would be that this would slow down any processes that the robot is engaged in. An example is given earlier in the paper of shelf stacking, which would be a repetitive industrial task where speed is prized.
> >
> > We agree that speed can also be useful. We chose to use slower speeds, due to the limitation in the data collection being 60Hz. We found that there could be large jumps of up to 16 degrees per timestep. This makes precise control of the stopping point difficult, as each timestep has a large change in position. If the speed of data collection was increased, we believe this could allow for a controller that is able to handle faster rotation. We have added some discussion on this into the Limitation and Future Work section.
> >
> > >I think a current limitation of the method is the reliance on the tape to normalise the frictional properties of the objects and would like to see this discussed in the limitations section.
> >
> > Thank you for this comment, this highlighted an improvement we needed to make to clarify  the use of tape. We do not use the tape to normalise the friction, rather, we use it to create multiple friction properties per object.

---

### Meta-Review · Area_Chair_3yHb · 2022-08-13

**Recommendation:** Accept (Poster)
**Confidence:** 4

**Metareview:**

**Strengths:**
•	The problem is relevant

•	The paper is well written with a clear presentation of its contributions.

•	Novel approach to pivoting task for in-hand manipulation.

**Weaknesses:**

•	Poor justification of the baseline model to compare the results of the proposed approach.

•	MLP design is not well justified.

•	Paper lacks many details in experimental sections.

•	Design choices are not well justified.

Post-rebuttal:
Please revise the paper according to the points discussed in the rebuttal phase for camera-ready submission.

**Best Paper Nomination:**

No

---

> ### Author Response · Authors · 2022-08-23
> **Author Response to Area Chair 3yHb**
>
> Thank you for your feedback. We have tried our best to update the manuscript according to the suggestions of the reviewers. The updated manuscript can be found at: https://openreview.net/pdf?id=sCyKrS0k5a9 (login required)
>
>
> >• Poor justification of the baseline model to compare the results of the proposed approach.
>
> We have added discussion into Section 6.1, further justifying the choice of our baseline. We believe this to be an appropriate baseline for our work, as the work by Chen (2021) serves as a recent approach to a similar task. Therefore, we think this approach serves as the most appropriate baseline. Furthermore, we have also trained an RNN to act as a simple recurrent model baseline, details of this study are shown in Section 9.11 of the supplementary material.
>
>
> >• MLP design is not well justified.
>
> We have added discussion about the MLP baseline model into Section 6.1. We have also added ablation studies on the size of the window into the supplementary material (Section 9.10).
>
>
> >• Paper lacks many details in experimental sections and design choices are not well justified.
>
> We have incorporated the changes pointed out by all the reviewers in regards to the experimental sections. If you have any further specific comments on where this could be improved, please let us know and we can improve the manuscript accordingly.